# Microencapsulated *Limosilactobacillus reuteri* Encoding Lactoferricin-Lactoferrampin Targeted Intestine against *Salmonella typhimurium* Infection

**DOI:** 10.3390/nu15245141

**Published:** 2023-12-18

**Authors:** Xueying Wang, Weichun Xie, Limeng Cai, Chuang Han, Hongdi Kuang, Yilan Shao, Senhao Zhang, Qi Zhang, Jiaxuan Li, Wen Cui, Yanping Jiang, Lijie Tang

**Affiliations:** 1College of Veterinary Medicine, Northeast Agricultural University, Harbin 150030, China; tycoon28644@163.com (X.W.); xieweichun_neau@163.com (W.X.); clmyoyo@163.com (L.C.); hanchuang3198@163.com (C.H.); 18246360750@163.com (H.K.); shaoyilan@neau.edu.cn (Y.S.); zhang_sen_hao@163.com (S.Z.); a502694221@foxmail.com (Q.Z.); lijiaxuan.1993@163.com (J.L.); cuiwen@neau.edu.cn (W.C.); 2Heilongjiang Key Laboratory for Animal Disease Control and Pharmaceutical Development, Northeast Agricultural University, Harbin 150030, China

**Keywords:** LFCA, *Salmonella typhimurium*, *Limosilactobacillus reuteri*, microcapsules, intestinal barrier

## Abstract

*Salmonella enterica* serovar Typhimurium (*S. typhimurium*) is an important foodborne pathogen that infects both humans and animals and develops acute gastroenteritis. As porcine intestines are relatively similar to the human ones due to their relatively similar sizes and structural similarity, *S. typhimurium* causes analogous symptoms in both. Novel strategies for controlling *S. typhimurium* infection are also desired, such as mucosal-targeted delivery of probiotics and antimicrobial peptides. The bovine lactoferricin-lactoferrampin-encoding *Limosilactobacillus reuteri* (LR-LFCA) strain improves intestinal barrier function by strengthening the intestinal barrier. Weaned piglets were selected for oral administration of microencapsulated LR-LFCA (microcapsules entrap LR-LFCA into gastro-resistant polymers) and then infected with *S. typhimurium* for 3 days. We found that orally administering microencapsulated LR-LFCA to weaned piglets attenuated *S. typhimurium*-induced production of inflammatory factors in the intestinal mucosa by inhibiting the nuclear factor-kappa B (NF-κB) and P38 mitogen-activated protein kinases (MAPK) signaling pathway. Moreover, microencapsulated LR-LFCA administration significantly suppressed the oxidative stress that may correlate with gut microbiota (reduced *Salmonella* population and increased α-diversity and *Lactobacillus* abundance) and intestinal function (membrane transport and metabolism). Our work demonstrated that microencapsulated LR-LFCA effectively targeted intestine delivery of *Lactobacillus* and antimicrobial peptides and modulated gut microbiota and mucosal immunity. This study reveals a novel targeting mucosal strategy against *S. typhimurium* infection.

## 1. Introduction

Among all foodborne pathogens, *Salmonella enterica* serovar Typhimurium (*S. typhimurium*) is the second most common serotype of *Salmonella* found in humans [1]. *S. typhimurium* is a gram-negative facultative intracellular bacterium that is an important cause of acute gastroenteritis outbreaks [2,3]. An inflamed gut can be treated with probiotics to modulate inflammation, enhance epithelial and mucosal barrier function, and alter gut microbiota composition [4,5]. There is a long history of using probiotics to treat gastrointestinal disorders, and they have to adapt to the various environmental changes found in the gastrointestinal tract [6]. An increasing interest is being shown in oral delivery of probiotics to the gut microbiome, but during intestinal colonization, probiotics are limited in their oral bioavailability due to the complexity of the environment within the gastrointestinal tract [7].

The World Health Organization states that a successful probiotic product must contain an adequate amount of viable microorganisms. A colonies per gram (CFU) or milliliter (mL) of 10^6^–10^7^ is essential for conferring health benefits [8]. Drying probiotic cultures for industrial purposes reduces transportation costs and storage requirements [9]. Spray drying has attracted much attention because of its high yield, speed, and ease [10,11]. A hot, dry air flow is applied to a continuous liquid film to convert it into droplets [12]. There is generally a 10 times reduction in energy consumption in spray drying compared to freeze drying [13]. Monosaccharides, polysaccharides, amino acids, proteins, and their combinations are frequently used as spray-drying agents [14]. Bacterial encapsulation is supported physically, and the protective material is resistant to adverse conditions in the gastrointestinal tract and maintains a bacteria’s original biological properties and viability [15].

In order for probiotics to exert their desired effects, it is important to take into account their nature. Short cationic molecules with amphipathic structures make up antimicrobial peptides, and infections caused by microbes are combated by them [16]. However, it is expensive and laborious to synthesize and purify these molecules. Pepsin-digested bovine lactoferrin is the source of lactoferricin (Lfcin B) and lactoferrampin (Lfampin), antimicrobial peptides [17]. There is a lot of evidence to suggest that *Limosilactobacillus reuteri* (*L. reuteri*) is effective in colonizing many mammals and strengthening their intestinal barrier [18,19]. For the regulation of mucosal immunity and the maintenance of mucosal integrity, Lfcin B and Lfampin (LFCA) fusion have been found to be effective in *Lactobacillus* or *Lactococcus* [20,21,22]. *Lactobacillus* and antimicrobial peptides can be delivered effectively via microencapsulated genetically engineered probiotic products, modulating mucosal immunity. Due to the comparable intestinal anatomy and microbial composition shared between pigs and humans, pigs serve as a valuable model for investigating the effects of probiotics on the human intestine and immunomodulation [23,24]. Biomedical research would benefit from an in-depth understanding of the pig gut microbiome [25]. Pig models are ideal for investigating probiotic supplementation effects and conducting translational studies for humans [26].

In this study, bovine lactoferricin-lactoferrampin-encoding *L. reuteri* (LR-LFCA) was coated with a protective agent to prepare probiotic products using spray drying, then we examined how they cope with harsh conditions. In addition, we evaluated the gut microbiota, intestinal biochemistry, and serum biochemistry effects of microencapsulated LR-LFCA on gut health. Based on our hypotheses, microencapsulated LR-LFCA would survive in an adverse environment, change the gut microbiota’s structure, and promote the intestinal health of humans and animals.

## 2. Materials and Methods

### 2.1. Bacterial Strains and Growth Conditions

Pig intestinal contents were used to obtain laboratory-isolated *L. reuteri* CO21. Detailed sequences of this strain’s 16S rRNA can be found at GenBank (accession number MK920155). Both LR-LFCA and LR-CON (the pPG612-EGFP plasmid was transformed into *L. reuteri* CO21) were engineered at Laboratory of Microbiology and Immunology, College of Veterinary Medicine, Northeast Agricultural University. We grew LR-LFCA and LR-CON at 37 °C for 16 h in MRS broth (Oxoid, Hampshire, UK). In LB broth (Oxoid, Hampshire, UK), *Staphylococcus aureus* (*S. aureus*) CVCC546, *S. typhimurium* SL1344, and *Escherichia coli* (*E. coli*) CVCC10141 were cultured for 12 h to reach 10^8^ CFU/mL.

### 2.2. Biological Characterization of LR-LFCA

Every 2 h, OD_600_ and colonies on MRS agar plates were measured to determine the growth curves. Our tests were performed using ELISA and western blotting to analyze cell culture supernatant and bacterial cell protein levels. Primary antibody used was anti-bovine lactoferrin (monoclonal antibody, prepared by our laboratory, obtained by immunizing BABL/c mice, and diluted at 1:200), and secondary antibody used was anti-mouse IgG HRP (Zhongshan Golden Bridge, Beijing, China, diluted at 1:5000). Recombinant proteins were quantitatively analyzed using a standard curve that was manually integrated with bovine lactoferrin.

### 2.3. LR-LFCA Antibacterial Activity Assays In Vitro

*S. aureus* CVCC546, *S. typhimurium* SL1344, and *E. coli* CVCC10141 were cultured in LB broth (OD_600_ 0.6–0.8) and then smeared onto LB agar plates (200 μL suspension). Cell-free supernatants (CFSs) were generated by filtering LR-LFCA and LR-CON supernatants (0.22 μm). After that, CFSs were added to the Oxford cup (200 μL) at its midpoint. The bacteriostatic ring was measured following 24 h of static cultivation at 37 °C. *S. aureus* CVCC546, *E. coli* CVCC10141, and *S. typhimurium* SL1344 were grown with LR-LFCA cell lysates (5.0 μg protein) or LR-CON cell lysates in LB medium at 37 °C for 24 h. PBS acted as negative control. Samples were observed with a transmission electron microscope (JEOL, JEM-1230, Tokyo, Japan) at 100 kV.

### 2.4. Preparation of Microencapsulated LR-LFCA

Precipitates from LR-LFCA cultures were centrifuged for 5 min at 3500 g after 16–18 h of culture at 37 °C. In the following step, the final product was spray dried after mixing the precipitates with the wall materials (skim milk 17.6%, sodium glutamate 18.55%, polyvinylpyrrolidone 25.5%, maltodextrin 22.1%, and gelatin 14.5%; all materials were purchased from Biotopped, Beijing, China). Spray dryer temperatures were 140 °C for the inlet and 60 °C for the outlet. As described previously [27], microcapsule surface morphology was examined using a field emission scanning electron microscope (JEOL, Tokyo, Japan) at a voltage of 5 kV.

### 2.5. Artificial Gastrointestinal Fluid and Bile Salt Resistance of Microencapsulated LR-LFCA

Testing was conducted on microencapsulated and nonencapsulated LR-LFCA for tolerance to artificial gastrointestinal fluid or bile salts (Solarbio, Beijing, China). At 37 °C, a 30, 60, 90, or 120-min incubation period was maintained with LR-LFCA in simulated gastric fluid (SGF; pH 2.0, pepsin, Biotopped, Beijing, China). Additionally, LR-LFCA was incubated for 1, 2, 3, or 4 h at 37 °C in simulated intestinal fluid (SIF; pH 7.5, trypsin, Biotopped, Beijing, China). We incubated LR-LFCA for 3 h in modified MRS broth containing 0.2%, 0.3%, 0.4%, and 0.5% bile salts. Then, LR-LFCA was cultivated on MRS agar plates for 24 h at 37 °C. Survival rate calculation (%): visible colony numbers in different environments/in MRS broth × 100%.

### 2.6. Animals and Experimental Design

The Animal Care and Use Committee of Northeast Agricultural University approved all experimental protocols (protocol code was NEAUEC20220358 and date of approval was 26 August 2022). Based on the NRC guidelines, the basal diet was formulated [28]. The feed formula is shown in Appendix A. During the study, all animal care and treatment methods were in accordance with the Laboratory Animal Management Regulations (revised 2016) of Heilongjiang Province, China.

In total, 28 weaned piglets (21-day-old Rong Chang pigs) were randomly assigned to three treatments with an average body weight of 5.65 ± 1.25 kg (balanced gender). The piglets were housed individually in cages made of stainless steel. The environmental conditions were carefully controlled, with a temperature of 30 ± 2 °C and a relative humidity range of 65–70%. From day 22 to day 49, the piglets received the following treatments: oral administration of (1) a basal diet (CON group; *n* = 12), (2) microencapsulated LR-LFCA (LR-LFCA group; 1.2 × 10^10^ CFU/kg, mixed into the basal diet; *n* = 8), (3) microencapsulated LR-CON (LR-CON group; 1.2 × 10^10^ CFU/kg, mixed into the basal diet; *n* = 8). Fourteen days post administration, 4 piglets were randomly chosen from the LR-LFCA and LR-CON groups and 8 piglets were randomly from CON group and given treatment as follows: (1) oral administration of *S. typhimurium* (6.0 × 10^11^ CFU dissolved in 10 mL PBS) to each piglet from day 36 to day 38 (ST+LR-LFCA group, *n* = 4; ST+LR-CON group, *n* = 4; ST group, 4 treatments from CON group, *n* = 4); (2) oral administration of saline only (CON group, 4 treatments from CON group, *n* = 4). After 7 days post-infection (at day 42), pigs were euthanized and samples were collected immediately. The average daily gain (ADG) was calculated by recording the body weight on every morning. The assigned scoring system for diarrhea was as follows: 0, normal; 1, loose stool; 2, loose/some diarrhea; 3, diarrhea; 4, severe, watery diarrhea. A diarrhea score of 2 or higher indicated the presence of diarrhea, while scores below 2 indicated the absence of diarrhea. The diarrhea rate was calculated using the formula diarrhea rate (%) = (number of piglets with diarrhea × number of days with diarrhea)/(number of piglets × total number of observational days) × 100, as described by Xie et al. [22].

### 2.7. Sample Collection and Organ Morphology Analysis

The sample collection scheme for the antibacterial activity of microencapsulated LR-LFCA against *S. typhimurium* was as follows: After being carefully collected, processed, and fixed in paraformaldehyde (Merck, Darmstadt, Germany), weaned piglets’ small intestines (middle duodenum, middle jejunum, and distal ileum), as well as other organs (liver, spleen, lung) were examined. Hematoxylin intestinal morphology was analyzed using hematoxylin and eosin (H&E) staining [29]. Photographs were taken with Leica Application Suite software (version 3.2.0, Leica, Wetzlar, Germany), using a Leica DM3000 microscope (Leica, Germany). Intestinal mucosal (duodenum, jejunum, ileum) and cecum contents were collected and stored at −80 °C. Blood samples taken from the anterior vena cava were placed in EDTA anticoagulant tubes and sterile tubes without additives (collected serum). Blood and sera were analyzed to determine the physiological and biochemical indices as described before [30].

### 2.8. Determination of the Immune Organ Index, Cytokines Levels and Intestinal Antioxidant Parameters

The weights of the collected organs were measured, and the weight indices (g/kg body weight) of the immune organs was calculated as follows: organ weight (g)/body weight (kg). Intestinal mucosal expression of interleukin (IL)-6, IL-4, IL-10, IL-1β, tumor necrosis factor-α (TNF-α), sIgA and serum levels of IgG were assessed using ELISA kits (Jiangsu Meimian Biological Technology Co., Ltd., Yancheng, China). In brief, double antibody sandwich method for measuring the absorbance of antibody–antigen–enzyme-labeled antibody complexes at 450 nm, based on standard curves, the protein content was calculated. ELISA kits were also used to determine antioxidant parameter (glutathione peroxidase; GSH-Px) levels in intestinal mucosa.

mRNA expression levels of myeloid differentiation factor 88 (MyD88), toll-interleukin-1 receptor domain-containing adapter protein inducing interferon-β (TRIF), TNF receptor-associated factor 6 (TRAF6), p38 mitogen-activated protein kinases (MAPK), activator protein-1 (AP-1), c-Fos, c-JUN, nuclear factor-kappa B (NF-κB) p50, inhibitor of NF-κB kinase (IKK)-α and IKK-γ were determined by real-time quantitative PCR. Jejunal and ileal mucosa RNA were extracted with an RNA Purification Kit (Sevenbio, Beijing, China). A comparative CT method (ΔΔCt method) was used to normalize results to GAPDH housekeeping genes. Detailed primer information can be found in Appendix A.

### 2.9. Microbiome Analysis

After extracting DNA from the cecum contents of the piglets, universal primers (shown in Appendix A) were used to amplify distinct regions of the 16S rRNA gene (V3–V4). The quantitation of PCR products was performed using a QUANTIFLUOR™ spectrofluorometer. Sequencing libraries were constructed from equal amounts of mixed, purified amplification products. Sequencing was completed at GENE DENOVO Co., Ltd., using the sequencing platform HiSeq2500 PE250. Our analysis was conducted using OmicShare tools, a free online platform that allows users to analyze sequencing data. Previously described methods were used to analyze the reads [16].

### 2.10. Statistical Analyses

Data were presented as mean results ± standard deviation (SD). A one-way ANOVA was used to analyze the statistical significance of the differences. Tukey’s post hoc analysis was then used to compare multiple groups. Significant *p*-values were reported as * *p* < 0.05, ** *p* < 0.01 or *** *p*< 0.001. Statistics were calculated using SPSS software (version 24.0, SPSS Inc., Chicago, IL, USA).

## 3. Results

### 3.1. Bioactivity of LR-LFCA and Stress Resistance of Microencapsulated LR-LFCA

Western blot analyses were performed to detect LFCA-E expression (Figure 1A). According to the analysis of the bacterial growth curves, LR-LFCA and LR-CON grow similarly (Appendix A). ELISA was used to calculate LFCA-E protein expression between 8 h and 24 h. In cell lysates and supernatants, the LFCA-E protein levels reached 1.54 mg/mg and 0.81 mg/mL, respectively, at 18 h (Figure 1B). Appendix A presents LR-LFCA’s antibacterial activity. The results showed different growth inhibition levels against *S. typhimurium* SL1344, *E. coli* CVCC10141 and *S. aureus* CVCC546. LR-LFCA cell lysates damaged the morphologies of *S. aureus* CVCC546, *E. coli* CVCC10141 and *S. typhimurium* SL1344, as revealed by negative staining and electron microscopy. As compared to untreated (control) bacteria, the LR-LFCA lysate-treated bacteria contained more atypical vesicles, protrusions, and filamentations (Figure 1C and Appendix A). Microcapsules with high survival rates were observed by scanning electron microscopy (SEM). Microcapsule particles showed irregular spherical structures with pits and folds on the surfaces that maintained structural integrity without rupture (Figure 1D). As a result of the rapid evaporation of water, microcapsules obtained by spray drying have this characteristic. Figure 1E illustrates the survival of free and microencapsulated LR-LFCA in porcine bile salts. Microencapsulated LR-LFCA showed better survival in porcine bile salts at concentrations ≤ 0.5% (*w*/*v*). Additionally, LR-LFCA encapsulated in microspheres tolerated simulated gastric and intestinal fluids well (Figure 1F and Appendix A).

### 3.2. Oral Administration of Microencapsulated LR-LFCA to Weaned Piglets Protected agsainst S. typhimurium Infection

Weaned piglets were orally administered microencapsulated LR-LFCA and LR-CON. Compared to the piglets in the CON group, microencapsulated LR-LFCA administration in early life significantly increased the body weight (*p* < 0.01) and average daily feed intake (*p* < 0.05) of piglets. The group LR-CON demonstrated no significant effect on average daily feed intake. No significant difference was observed in the feed-to-gain ratio among groups, but it was smaller in the LR-LFCA group than in the LR-CON and CON groups (Appendix A). Compared to the piglets in the CON group, microencapsulated LR-LFCA administration in early life significantly increased the liver coefficient and immune organ indices of piglets (*p* < 0.01; Appendix A). The weaned piglets received continuous oral microencapsulated LR-LFCA for an average of 21 days. We then analyzed their blood and serum to measure physiological and biochemical parameters. The physiological indices of the piglets fell within the normal range; hence, our study showed no adverse health effects in association with the treatments (Appendix A). Notably, no significant disparities were observed in kidney and liver function, and the levels of blood lipids and glucose markers (*p* > 0.05) remained within the established normal range. Moreover, there was an increase in the total proteins and albumin in the LR-LFCA group (Appendix A).

Weaned pigs were administered oral microencapsulated LR-LFCA and then infected with *S. typhimurium* for 3 days and sacrificed at day 7 post infection. Compared to the piglets in the ST and ST+LR-CON groups, the piglets in the ST+LR-LFCA group exhibited lower diarrhea incidence (Figure 2A). The effects of administering microencapsulated LR-LFCA on the bodyweight of weaned piglets challenged with *S. typhimurium* are presented in Figure 2B, where a significantly lower weight loss is observed in the ST+LR-LFCA group. The ST+LR-LFCA group piglets showed increased liver coefficients and immune organ indices compared to those in the ST group (*p* < 0.05; Appendix A). In order to determine the host immune responses, we quantified the IgM, IgA, IgG concentrations in the serum of the piglets (Figure 2C–E). Compared to those in the ST group, the piglets in the ST+LR-LFCA group showed higher IgM, IgA, and IgG levels in their serum. However, there was no apparent difference in the levels of these parameters between the ST and ST+LR-CON groups. We observed an increase in the GSH-Px levels in the ST+LR-LFCA group compared to that in the ST or ST+LR-CON groups (Figure 2F). The levels of sIgA in the duodenum, jejunum, and ileum increased significantly in the ST+LR-LFCA group (Figure 2G). We then analyzed their blood and serum to assess certain physiological and biochemical parameters. The numbers of leukocytes and neutrophils decreased significantly in the *S. typhimurium*-treated piglets compared to those in the ST group (*p* < 0.01). No significant differences in the levels of neutrophils were detected between the ST+LR-CON and ST group (*p* > 0.05, Appendix A). There was an increase in the globulin content in the ST+LR-LFCA group, but the elevation was not statistically significant (Appendix A). Compared to the ST group, the total serum protein content in the ST+LR-LFCA group increased significantly (*p* < 0.01). The weaned piglets treated with microencapsulated LR-LFCA showed a trend of increased triglyceride content (Appendix A). Histopathological analysis of tissue sections revealed that infection with *S. typhimurium* led to inflammation and intestinal injury in the piglets’ jejunum and ileum (weaned piglets infected with *S. typhimurium*) compared to those in the ST group. When microencapsulated LR-LFCA was administered to weaned piglets, it reduced the infiltration of inflammatory cells induced by *S. typhimurium* and improved the intestinal mucosal structure (Figure 2H). Marked infiltration of inflammatory cells and hemorrhage were seen in the ST and ST+LR-CON groups. Both the ST and ST+LR-CON groups exhibited higher degrees of pulmonary inflammatory cell infiltration and hemorrhaging than the ST+LR-LFCA group. The spleens of the piglets in the ST+LR-CON and ST groups exhibited slight to severe degrees of necrosis in splenic lymphocytes and congestion. The livers of treated weaned piglets presented variable degrees of hepatocyte degeneration after the *S. typhimurium* challenge. However, microencapsulated LR-LFCA treatment significantly alleviated these symptoms compared to those in the ST group (Appendix A).

### 3.3. Oral Administration of Microencapsulated LR-LFCA to S. typhimurium-Infected Weaned Piglets Improved Gut Microbiota Composition

We assessed the cecal microbiota composition of piglets post *S. typhimurium* infection. The alpha diversity indices are displayed in Figure 3A. A sufficient sequencing depth was confirmed through the rarefaction curves (Figure 3B). The abundance-based coverage estimator, Chao1, Sob, and Shannon indexes were higher in the ST+LR-LFCA group than in the ST and ST+LR-CON groups. Dendrograms of the hierarchical cluster analysis and heatmaps of the sample-to-sample distances were obtained through beta diversity analysis (Figure 3C,D). The CON group might be functionally closer to ST+LR-LFCA than ST+LR-CON or ST. Phylum, and the genus level comparison of cecal microbiota abundance in piglets is provided in Figure 3E–G. The most abundant phyla in piglets were Bacteroidetes and Firmicutes. Proteobacteria and Actinobacteria followed.

We compared the differential enrichment of taxa in the microbiomes using linear discriminant analysis (LEfSe, LDA > 3.3). The relative abundance of *Lactobacillus*, Prevotellaceae, and Ruminococcaceae in the CON group was significantly higher than that in the ST group at the genus level. By contrast, the relative abundance of *Salmonella*, Clostridiales, Christensenellaceae, and *Eubacterium* in the ST group was significantly higher than that in the CON group (Figure 4A and Appendix A). A more comprehensive discussion of these data is presented in Appendix A. When comparing the ST and ST+LR-LFCA groups, *Lactobacillus*, *Mitsuokella*, and Veillonellaceae displayed a relatively higher abundance in the ST group at the genus level. However, the relative abundance of *Salmonella*, Ruminococcaceae, *Clostridiaceae*, and Christensenellaceae in the ST group was significantly higher than that in the ST+LR-LFCA group (Figure 4B and Appendix A). Details are described in Appendix A. The administration of *S. typhimurium* resulted in an imbalanced microflora in the gut, whereas the administration of microencapsulated LR-LFCA restored the imbalanced gut flora and modulated the intestinal microecology. Using Tax4Fun, the Kyoto Encyclopedia of Genes and Genomes (KEGG) and KEGG orthology pathways were predicted to investigate the changes in the gut microbial functional profiles based on 16S rRNA gene sequencing data. The expression of pathways related to membrane transport were much higher in the piglets in the ST+LR-LFCA group than in the other groups (Figure 4C). The piglets showed a significant decrease in glycan biosynthesis and metabolism after *S. typhimurium* administration, whereas microencapsulated LR-LFCA administration significantly alleviated this change. The piglets in the ST group showed signs of signal transduction, endocrine system, and biosynthesis of other secondary metabolite function abnormalities. However, this change was partially mitigated by microencapsulated LR-LFCA administration (Figure 4D).

### 3.4. Oral Administration of Microencapsulated LR-LFCA to Weaned Piglets Protected against Changes in S. typhimurium-Induced Inflammatory Factor Levels in the Intestinal Mucosa

LR-LFCA modulated *S. typhimurium*-treated piglets by the upregulation of proinflammatory factors (TNF-α, IL-6, IL-1β) and the downregulation of anti-inflammatory factors (IL-10, IL-4) in the jejunum and ileum. Microencapsulated LR-LFCA administration significantly reduced the IL-1β, IL-6 and TNF-α levels in the ST+LR-LFCA group (Figure 5A and Appendix A). The levels of IL-1β and IL-6 were not different between the ST+LR-CON group and the ST group. Furthermore, administration of microencapsulated LR-LFCA also upregulated the IL-4 and IL-10 levels (Figure 5B and Appendix A). This result indicates that microencapsulated LR-LFCA downregulated proinflammatory factors and increased anti-inflammatory factors following oral administration in *Salmonella*-infected piglets. We further assessed the MyD88 and TRIF, and a significant reduction in MyD88 and TRIF expression levels was observed in the ST+LR-LFCA group following microencapsulated LR-LFCA administration (Figure 5C and Appendix A). To investigate this finding further, we next examined the TRAF6, p38 MAPK, c-Fos, c-JUN, IKK-α, IKK-γ, NF-κB p50 and AP-1 relative mRNA expression as measured in triplicate using real-time PCR (Figure 5D and Appendix A). It was shown that weaned piglets with oral administration of microencapsulated LR-LFCA may experience enhanced anti-inflammatory abilities through inhibition of the p38 MAPK and NF-κB pathway. The corresponding mechanism is shown schematically in Figure 5E.

## 4. Discussion

During *L. reuteri* colonization, wall materials may provide a barrier against external pressures. We found that composite wall materials (gelatin, maltodextrin, sodium glutamate, skim milk and polyvinylpyrrolidone) had an excellent protective effect on LR-LFCA. Gelatin shows excellent film-forming ability [31]. Skim milk proteins and maltodextrin protect bacteria from the damage caused by harsh environments [32,33]. As reported by Bustamante et al. [34], during storage, Bifidobacterium and *Lactobacillus* plantarum encapsulated with spray-dried composite carriers survived at 98%. Here, we highlighted that the microencapsulated LR-LFCA was highly stress-resistant. The microencapsulation of probiotics may protect them from the gastrointestinal environment. Given these findings, LR-LFCA can be isolated from the external environment by wall materials, which prolongs LR-LFCA survival.

*L. reuteri* colonizes piglet intestinal mucosa and improves immune and antioxidant functions and intestinal microbiota, which in turn boosts growth, decreases morbidity, and improves survival [35,36]. Researchers in the past have shown that LFCA expressed by recombinant *Lactococcus lactis* promotes piglet growth and immunity and alleviates colon damage induced by DSS in mice [20,21]. According to our data, microencapsulated LR-LFCA increased body weight and daily food intake significantly. Microencapsulated LR-LFCA administration in early life significantly increased the liver coefficient and immune organ indices of piglets. These data revealed that the microencapsulated LR-LFCA improved the immunity of piglets. In addition to providing information about an animal’s metabolism, serum biochemical parameters can also provide important information about their health [37]. Porcine serum protein (or serum albumin), which is synthesized by the liver, represents a rich source of body protein and serves as an indicator of protein reserves [38,39]. Porcine serum protein can also repair tissue and provide energy. After transforming into plasma cells, B cells secrete globulin, reflecting the body’s resistance [40]. It was shown in our study that LR-LFCA microcapsules administered orally to weaned piglets improved serum albumin levels and total protein levels.

In the body, immune factors such as IgM, IgA, and IgG play a critical defense role [41]. sIgA is paramount to mucosal immunity and provides extended protection [42]. There was a significant increase in serum IgM, IgA, and IgG levels following microencapsulated LR-LFCA administration, as well as intestinal sIgA levels after LR-LFCA administration. Based on these data, it was discovered that microencapsulated LR-LFCA improved humoral immunity in piglets. Generally, mucosal barrier function and intestinal morphology are closely related [43]. Previous research has shown that pathogenic infections increase intestinal permeability and damage intestinal morphology, leading to pathology [44]. However, the probiotic *Pediococcus acidilactici* can restore intestinal morphology and enteric immunity [45]. In our study, microencapsulated LR-LFCA had a beneficial effect on damaged intestinal villi in the ileum and jejunum; as a result, intestinal integrity and mucosal barrier function were improved in group ST+LR-LFCA.

The pathogenic gram-negative bacterium *S. typhimurium* has great clinical significance and has emerged as a pervasive trigger for enteritis in human and nonhuman hosts [46]. Due to the anatomical, physiological, immunological and susceptibility to pathogen similarities between human and porcine intestines [24,47,48,49], the piglet model was used in the research regarding salmonellosis and the gastrointestinal system. According to previous studies, *S. typhimurium* infection is associated with intestinal inflammation and cellular injury [50,51]. Our results are consistent with this finding. The *S. typhimurium* challenge upregulated inflammatory factors and downregulated anti-proinflammatory factors. However, LR-LFCA microcapsules alleviated this damage. There was a significant upregulation of levels of IL-4 and IL-10 after microencapsulated LR-LFCA administration, as well as a significant reduction in the levels of IL-6, TNF-α and IL-1β. This is consistent with the results from previous studies [52,53]. The administration of certain *Lactobacillus* and antimicrobial peptides can reduce *Salmonella* infection by modulating cytokine gene expression. In addition, microencapsulated LR-LFCA was found to suppress transcriptional activation of inflammatory genes by p38 MAPK and NF-κB, which may explain its anti-inflammatory effects. It will be interesting to see whether the effect of *S. typhimurium* infection is mediated by any toxin or structure and reversed by LR-LFCA.

Previous studies have shown that it is important for humans or animals to maintain a healthy intestinal microbiota, and high-throughput sequencing provides insights into microbial populations [54,55]. This study compared the diversity and composition of microbiota in *S. typhimurium*-infected weaned piglets after treatment with LR-LFCA microcapsules using high-throughput sequencing. Three groups of weaned piglets were randomly divided and supplemented with LR-LFCA microcapsules, LR-CON microcapsules, or saline for 14 days, followed by the oral gavage of *S. typhimurium*. Thereafter, we compared the gut microbiota among the groups, and diversity and composition of the microbiota in the cecum differed among the three treatments.

Intestinal microflora could be another confounding factor, especially in mucosal immunity [56]. According to Shannon indices, the alpha diversity of gut microbiota was enhanced by LR-LFCA microcapsules when administered orally. Moreover, based on sample-to-sample distances and hierarchical cluster analysis, the gut microbial compositions differed significantly between the groups. A comparison with the CON group also revealed an increase in the proportion of *Bacteroides* and Proteobacteria among weaned pigs who had been administered *S. typhimurium* orally. After oral administration of LR-LFCA microcapsules, there was a significant reduction of *Salmonella*, Ruminococcaceae_UCG-005 and *Phascolarctobacterium* abundance, while the *Veillonella*, *Lactobacillus*, and *Mitsuokella* proportions increased. In these studies, oral administration of microencapsulated LR-LFCA to weaned piglets infected with *S. typhimurium* was found to promote the growth of beneficial bacteria while impeding the growth of harmful pathogens. This manipulation of gut microbiota has the potential to safeguard against *S. typhimurium* infection and ultimately reinstate the proper functioning of the intestinal mucosal barrier by regulating mucosal oxidative stress and maintaining a balanced pro- and anti-inflammatory response. The main limitation of our study was the lack of piglets in the synthetic oral LFCA alone group (limited by cost).

## 5. Conclusions

In conclusion, LR-LFCA was microencapsulated using a mixture of wall materials (gelatin, maltodextrin, polyvinylpyrrolidone, sodium glutamate, and skim milk) in this study, which provided in-depth analysis. Stress resistance was increased by microencapsulating LR-LFCA. This study also established that microencapsulated LR-LFCA could play an essential role via host defense against pathogens and maintenance of mucosal barrier integrity in weaned piglets. Thus, microencapsulated LR-LFCA is critical in the regulation of intestinal mucosal homeostasis, barrier function and improving the anti-inflammatory ability of *S. typhimurium*-infected piglets. The current results provide a theoretical basis for applying microencapsulated LR-LFCA as a microecological agent that regulates mucosal immunity.

## Figures and Tables

**Figure 1 nutrients-15-05141-f001:**
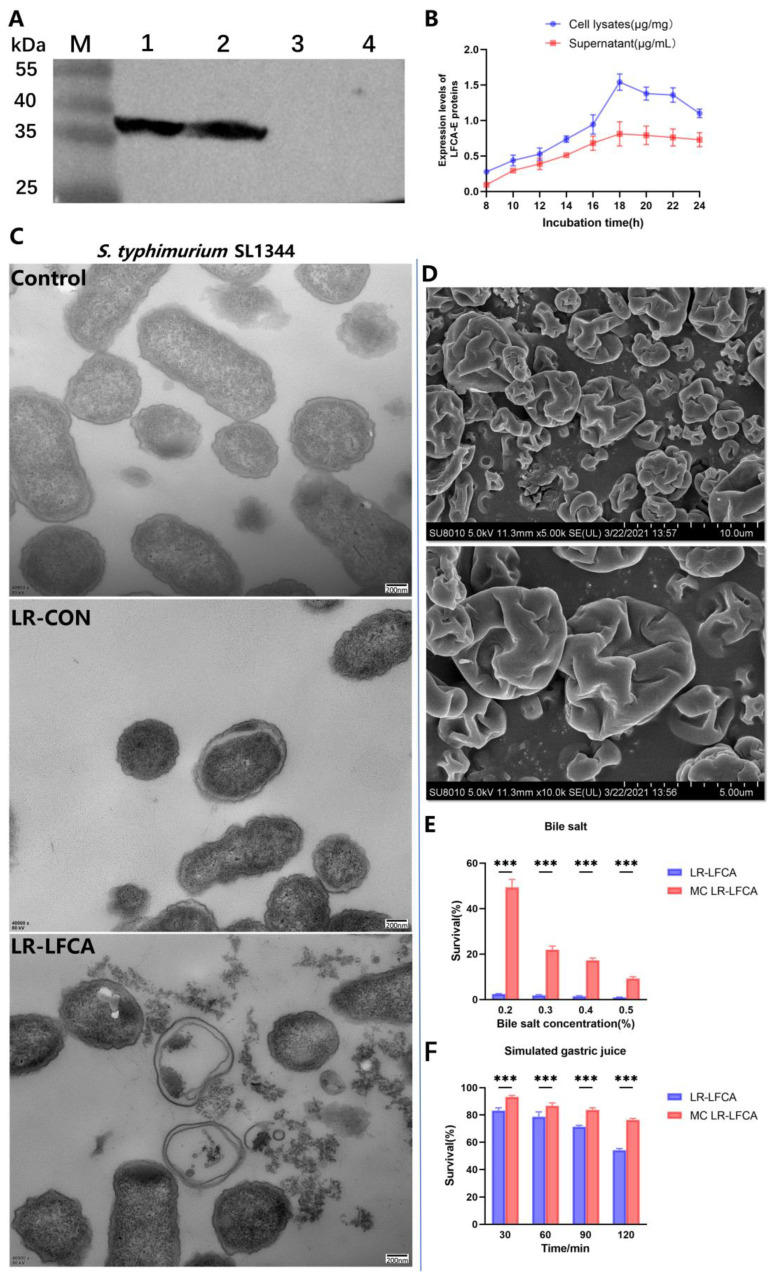
Protein expression, antimicrobial activity of LR-LFCA and stress resistance of microencapsulated LR-LFCA. (**A**) LFCA-E protein expression was analyzed using western blotting, which resolved an immunoreactive band at 38 kDa. LR-LFCA and LR-CON were cultured for 24 h, and the LFCA-E protein expression levels were assayed in the cell lysates and culture supernatant concentrated 50 fold using trichloroacetic acid (TCA) precipitation. 1: The cell lysates of LR-LFCA. 2: The culture supernatant of LR-LFCA. 3: The culture supernatant of LR-CON. 4: The cell lysates of LR-CON. Irrelevant lanes were omitted. (**B**) Expression levels of LFCA-E proteins in the cell lysates and supernatants. (**C**) Ultrastructural damage in *S. typhimurium* SL1344 treated with cell lysates (5.0 µg protein) from LR-LFCA and LR-CON. Control, *S. typhimurium* SL1344 treated with PBS; LR-CON, *S. typhimurium* SL1344 treated with cell lysates from LR-CON; LR-LFCA, *S. typhimurium* SL1344 treated with cell lysates from LR-LFCA. Cells were analyzed by electron microscopy. (**D**) Scanning electron microscopy (SEM) images of microcapsules. (**E**) Survival rate of microencapsulated LR-LFCA in the presence of different bile salts concentrations. (**F**) Survival of microencapsulated LR-LFCA in simulated gastric juice. *** *p* < 0.001.

**Figure 2 nutrients-15-05141-f002:**
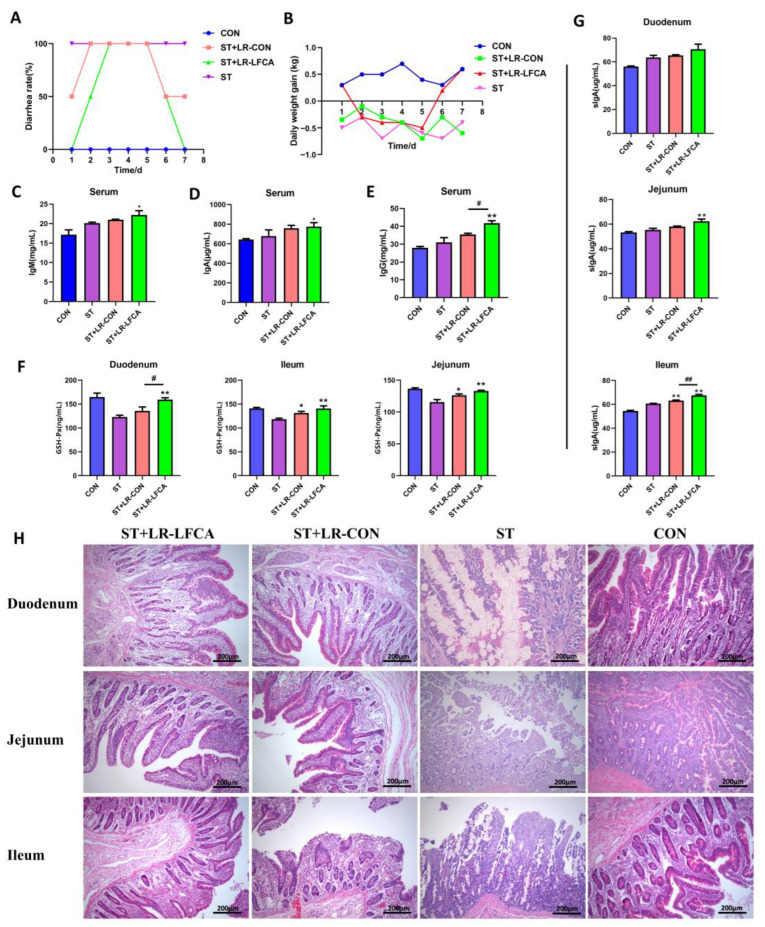
Microencapsulated LR-LFCA administration effects on clinical symptoms, pathological changes, immunoglobulin, antioxidant levels in *S. typhimurium*-infected piglets. (**A**) The incidence of diarrhea (%) was calculated as the [(number of piglets with diarrhea × number of days of diarrhea)/(total number of experiment piglets × number of days of the whole experiment)] × 100%. (**B**) Daily weight gain of piglets (kg) after *S. typhimurium* infection. (**C**–**G**) Blood and sera were obtained after *S. typhimurium* infection. IgA, IgG, and IgM concentrations in the serum, as well as glutathione peroxidase (GSH-Px) and sIgA levels in the intestinal mucosa were measured using ELISA. (**H**) Intestinal morphology and pathology in the duodenum, jejunum, and ileum of infected piglets, as observed using H&E staining. Data are presented as the mean ± SD. * *p* < 0.05 vs. ST; ** *p* < 0.01 vs. ST; # *p* < 0.05 vs. ST+LR-LFCA; ## *p* < 0.01 vs. ST+LR-LFCA.

**Figure 3 nutrients-15-05141-f003:**
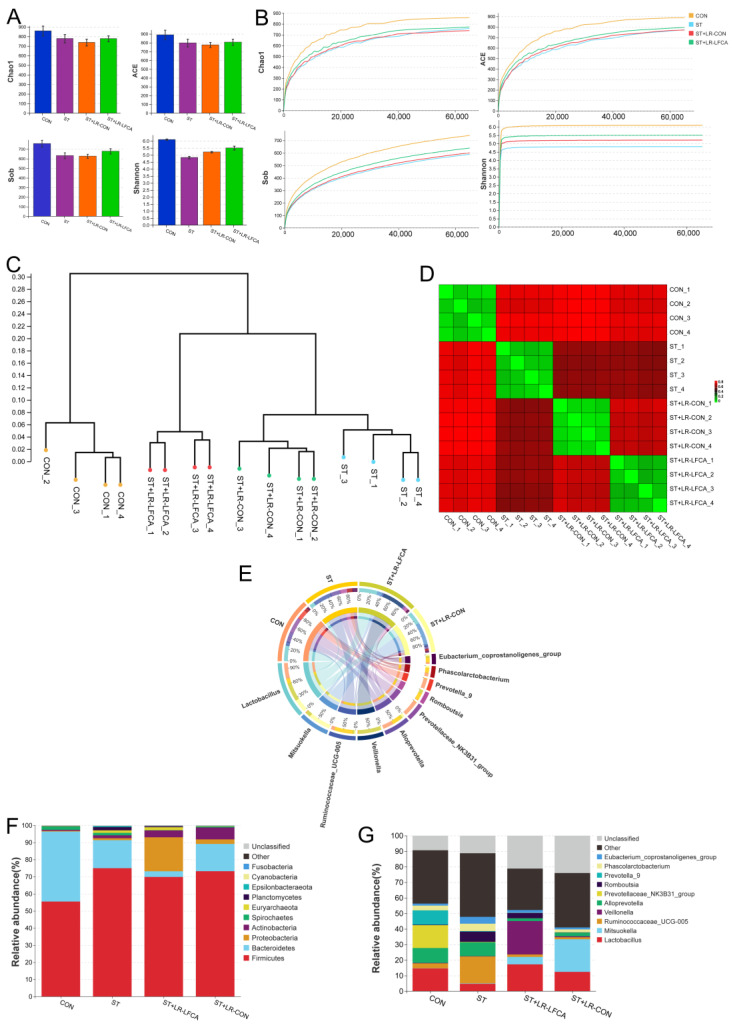
The cecal microbiota composition of piglets after being challenged with *S. typhimurium*. (**A**) Chao1 index, Ace index, Sob index, Shannon index. (**B**) Alpha rarefaction curve. (**C**) Dendrogram showing the cluster analysis (unweighted pair group method with arithmetic mean [UPGMA]) of samples. (**D**) Heatmap of sample-to-sample distances, showing hierarchical clustering of sample-to-sample distances. (**E**) Circos diagram illustrating genus-level microbial compositions in samples. (**F**) Relative abundance of the cecal microbiota at the phylum level and LefSe analysis of the dominant biomarker taxa between groups. The log10 LDA score threshold was set to 3.3. (**G**) Relative abundance at the genus level in cecal microbiota.

**Figure 4 nutrients-15-05141-f004:**
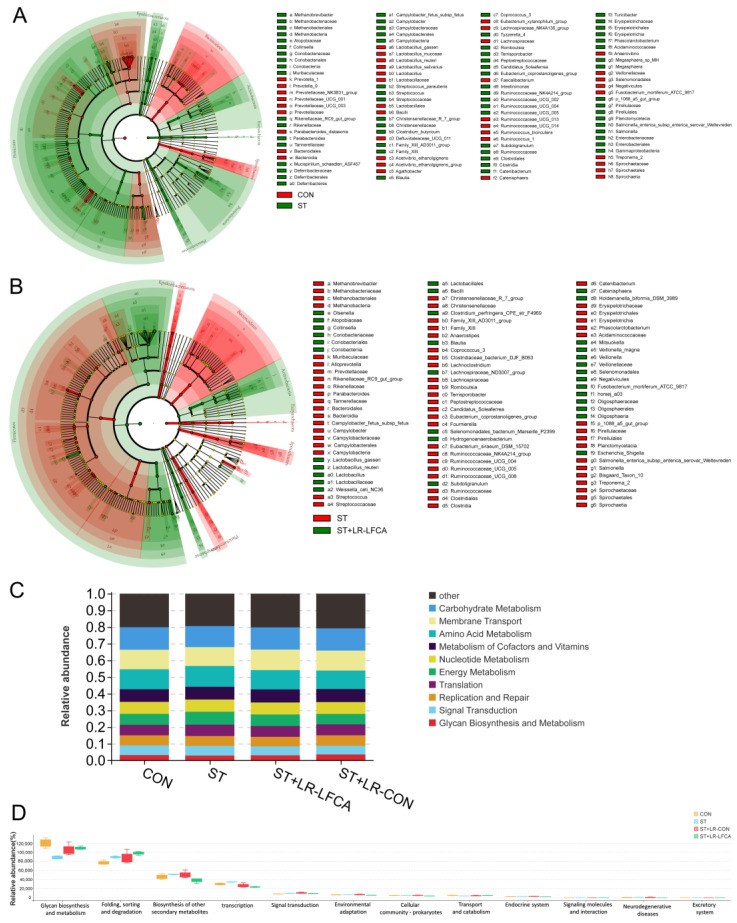
Changes in intestinal microbial diversity in *S. typhimurium*-infected piglets after microencapsulated LR-LFCA administration. (**A**) Differences between the CON and ST groups. (**B**) Differences between the ST and ST+LR-LFCA groups. (**C**) Functional profiling of gut microbiomes based on PICRUSt2 analyses of 16S data. (**D**) Functional prediction based on the KEGG database was annotated based on 16S rRNA sequence data.

**Figure 5 nutrients-15-05141-f005:**
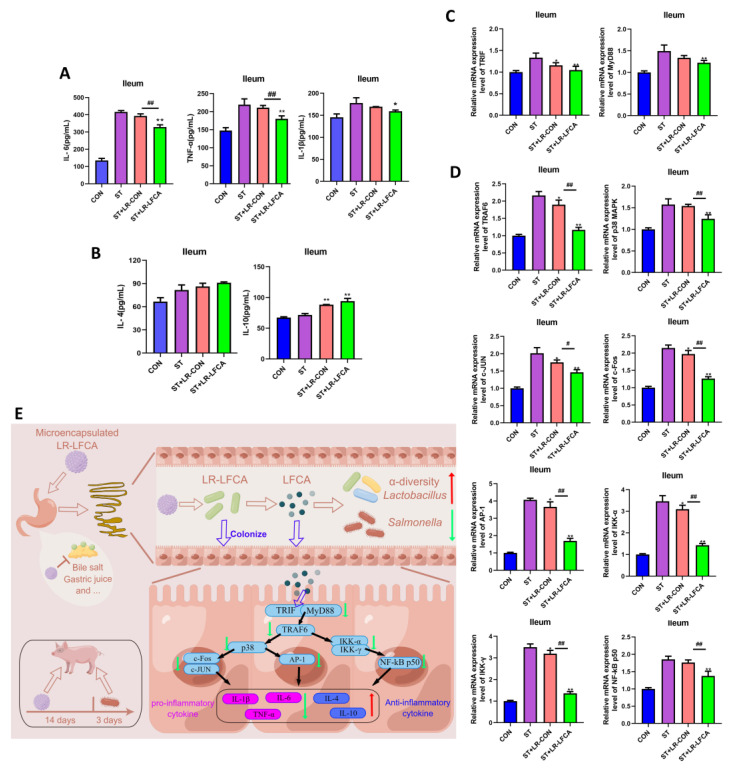
Impact of *S. typhimurium* infection on inflammatory factors in ileum mucosa post microencapsulated LR-LFCA administration. (**A**) IL-6, TNF-α, IL-1β, (**B**) IL-4, and IL-10 protein levels in the ileum as measured in triplicate using ELISA. (**C**) TRIF, MyD88, (**D**)TRAF6, p38 MAPK, c-JUN, c-Fos, AP-1, IKK-α, IKK-γ, and NF-κB p50 relative mRNA expression in the ileum as measured in triplicate using real-time PCR. (**E**) Microencapsulated LR-LFCA regulates intestinal mucosal immune mechanism (Red arrows represent elevated levels, and green arrows designate reduced levels compared to ST group). Data are presented as the mean ± SD. * *p*< 0.05 vs. ST; ** *p*< 0.01 vs. ST; # *p*< 0.05 vs. ST+LR-LFCA; ## *p* < 0.01 vs. ST+LR-LFCA.

## Data Availability

All data generated or analyzed during this study are included in this publication (and the Appendix A). All 16S rRNA sequencing data were submitted under accession No. PRJNA795599.

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
