# Peer review of "Microencapsulated Limosilactobacillus reuteri Encoding Lactoferricin-Lactoferrampin Targeted Intestine against Salmonella typhimurium Infection"

_nutrients, 2023, doi:10.3390/nu15245141_

Round 1
Reviewer 1 Report
Comments and Suggestions for Authors
Review - manuscript
Microencapsulation of LimosiLactobacillus reuteri Encoding Bovine Lactoferrin Peptides Enhances Mucosal Immunity against Salmonella enterica serovar Typhimurium SL1344 Infection
Dear authors, this is an exciting manuscript. However, I just cant understand the Figures presented in the Results section.
OVERALL COMMENTS:
In the study, “bovine lactoferricin-lactoferrampin-encoding L. reuteri (LR-LFCA) was coated with a protective agent to prepare probiotic products using spray drying… examined how they cope with harsh conditions”. In addition, the authors “evaluated gut microbiota, intestinal biochemistry, and serum biochemistry effects of microencapsulated LR-LFCA on gut health”.
TITLE
The proposed title is too long. I suggest authors try to reduce it (please try to correlate the title to the objective (page 2, lines 74-79) more clearly).
ABSTRACT
Please define all the abbreviations in this section. They should be defined the first time they are mentioned in the text.
KEYWORDS
I suggest removing the definition for lactoferricin-lactoferrampin. I also suggest adding “intestinal barrier”.
INTRODUCTION
In the Introduction, only references published up to 2021 are found. I think it is interesting to include references published in 2022-2023, showing the evolution of the subject studied. If no suitable publications exist in these years, the authors should inform this issue. Many interesting papers can be found at PUBMED.com.
There are sentences in this section that have no references. Please include in the following sentences: page 2, lines 44-46, and 71-72.
Many abbreviations need to be defined along with the text. As examples, I can cite TNF-α, IL, MAPK, NF-κB,
METHODS
In line 122-123 we can read “The Animal Care and Use Committee of Northeast Agricultural University approved 130 all experimental protocols (protocol code was NEAUEC20220358)”. Please include the month and the year of the approval.
RESULTS
This section is difficult to understand; the quality of the images is very poor. Please provide better-quality images. For these reasons, reading the words the in Figures 1- 2, and 5 is impossible. Figure 3 and 4 is even worse. For these reasons, I suggest describing the results of taking some time to improve images and sending them back to this reviewer.
LIMITATIONS
Please include the limitations of this study.
REFERENCES
As pointed out above, I suggest including more references published in 2023 along with the text.
Comments on the Quality of English LanguageMinor revisions.
Reviewer 2 Report
Comments and Suggestions for Authors
Wang et al. submitted article entitled Microencapsulation of LimosiLactobacillus reuteri Encoding Bovine Lactoferrin Peptides Enhances Mucosal Immunity against Salmonella enterica serovar Typhimurium SL1344 Infection. The work may be interesting for the readers. I just have a small comment.The authors acknowledge the support of HOME for Researchers and FigDraw for Figure 6E drawing materials. However, I do not see Fig. 6 in the manuscript. Drawings need to be corrected - contain too much information, the resolution is very poor, and font illegible - may divide these drawings and include some of the data in the supplement as additional drawings.
Round 2
Reviewer 1 Report
Comments and Suggestions for Authors
Dear authors,
Thank you for correcting the ms according to my suggestions.
With regards.
minor